# A Phase II trial of Higher RadiOtherapy Dose In The Eradication of early rectal cancer (APHRODITE): protocol for a multicentre, open-label randomised controlled trial

Eleanor M Hudson ,[1] Samantha Noutch,[1] Sarah Brown ,[1] Ravi Adapala,[2]
Simon P Bach,[3] Carole Burnett,[4] Alwyn Burrage,[5] Alexandra Gilbert,[6]
Maria Hawkins ,[7] Debra Howard,[8] Monica Jefford,[9] Rohit Kochhar,[10]
Mark Saunders,[11] Jenny Seligmann,[6] Alexandra Smith,[1] Mark Teo,[4]
Edward JD Webb ,[12] Amanda Webster,[8] Nicholas West,[6]
David Sebag-Montefiore,[6] Simon Gollins,[13] Ane L Appelt[6]

SG and ALA are joint senior authors.

For numbered affiliations see end of article.

**Correspondence to**
Dr Ane L Appelt;
A.L.Appelt@leeds.ac.uk

## ABSTRACT

**Introduction** The standard of care for patients with localised rectal cancer is radical surgery, often combined with preoperative neoadjuvant (chemo)radiotherapy. While oncologically effective, this treatment strategy is associated with operative mortality risks, significant morbidity and stoma formation. An alternative approach is chemoradiotherapy to try to achieve a sustained clinical complete response (cCR). This non-surgical management can be attractive, particularly for patients at high risk of surgical complications. Modern radiotherapy techniques allow increased treatment conformality, enabling increased radiation dose to the tumour while reducing dose to normal tissue. The objective of this trial is to assess if radiotherapy dose escalation increases the cCR rate, with acceptable toxicity, for treatment of patients with early rectal cancer unsuitable for radical surgery.

**Methods and analysis** APHRODITE (A Phase II trial of Higher RadiOtherapy Dose In The Eradication of early rectal cancer) is a multicentre, open-label randomised controlled phase II trial aiming to recruit 104 participants from 10 to 12 UK sites. Participants will be allocated with a 2:1 ratio of intervention:control. The intervention is escalated dose radiotherapy (62 Gy to primary tumour, 50.4 Gy to surrounding mesorectum in 28 fractions) using simultaneous integrated boost. The control arm will receive 50.4 Gy to the primary tumour and surrounding mesorectum. Both arms will use intensity-modulated radiotherapy and daily image guidance, combined with concurrent chemotherapy (capecitabine, 5-fluorouracil/leucovorin or omitted). The primary endpoint is the proportion of participants with cCR at 6 months after start of treatment. Secondary outcomes include early and late toxicities, time to stoma formation, overall survival and patient-reported outcomes (European Organisation for Research and Treatment of Cancer (EORTC) Quality of Life Questionnaires QLQ-C30 and QLQ-CR29, low anterior resection syndrome (LARS) questionnaire).

### Strengths and limitations of this study

► A Phase II Trial of Higher Radiotherapy Dose in the Eradication of Early Rectal Cancer is a prospective, multicentre randomised controlled trial, in contrast to previous studies of organ preservation that have primarily been retrospective, single centre or single arm.

► The trial offers treatment for patients who are not good candidates for (standard) surgical management, thus serving a group of patients often excluded from clinical trials.

► Long-course radiotherapy may be intolerable for some very frail patients, for whom short-course treatment may be preferential; this study will not assess short-course radiotherapy.

► The phase II design is focused on generating preliminary evidence of efficacy; a subsequent phase III trial will be required to evaluate longer term clinical impact.

► The study has embedded a strong focus on patients and their assessment of outcomes, including patient-reported outcome measures and a translational substudy of patient treatment preferences.

**Ethics and dissemination** The trial obtained ethical approval from North West Greater Manchester East Research Ethics Committee (reference number 19/NW/0565) and is funded by Yorkshire Cancer Research. The final trial results will be published in peer-reviewed journals and adhere to International Committee of Medical Journal Editors guidelines.
**Trial registration number** ISRCTN16158514.

## INTRODUCTION

Colorectal cancer is the third most common cancer in the UK. Each year, around 42 000

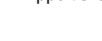

new cases are diagnosed, with over one-quarter within the rectum.[1] The standard of care for patients with localised rectal cancer is radical surgery combined with a selective use of preoperative radiotherapy and adjuvant chemotherapy. Radical surgery consists of total mesorectal excision (TME) which is an oncologically effective treatment for early stage rectal cancer; only 2% and 12% of patients will experience local or distant failure, respectively.[2–5] However, radical surgery is associated with risks of operative mortality, significant morbidity and the frequent need for stoma.

Mortality following elective major colorectal cancer surgery has fallen significantly in recent years and the 2020 UK National Bowel Cancer Audit Annual Report, relating to 2018–2019, shows overall 90-day mortality of 3%.[6] However, postoperative mortality for rectal cancer increases with age as demonstrated in the Dutch TME trial[7 8] and is highly dependent on patients' general physical fitness, with significantly increased 30-day mortality in patients with American Society of Anesthesiologists grade III and above physical status.[9]

In the 2020 UK National Bowel Cancer Audit,[6] 86% of patients with rectal cancer had a stoma following major resection, including 78% of patients undergoing anterior resection. Eighteen months later, 29% of these patients still had a stoma. Stoma-related morbidity affects at least 50% of patients; with the most common problems including high-output stomas, stoma prolapses, small bowel obstruction and wound infections.[10] Among the patients not undergoing radical resection, many are older or frailer, with comorbidities, or those who may struggle to manage a stoma. Surgery is often not a good option for these patients yet there is no standard alternative treatment.

An alternative approach to surgery is the use of radiotherapy with or without chemotherapy (chemoradiotherapy, CRT) to try to achieve a sustained clinical complete response (cCR) and potential long-term tumour control. Registration studies of patients with (primarily) locally advanced cancer who have achieved a cCR after neoadjuvant treatment, and opted for non-surgical management, have reported on the oncological safety of this approach.[11–13] A systematic review of 23 cohort studies of patients managed non-surgically following cCR suggested a pooled 2-year local regrowth rate of 21.3% in 575 cCR patients. The vast majority (93.2%) of patients with local regrowth were subsequently managed with salvage surgery.[14] Consequently, a policy of 'active surveillance' is increasingly being adopted for patients in whom post-CRT endoscopy and MRI suggest a cCR.

Most published studies have focused on an 'opportunistic' approach in patients achieving a cCR following CRT as part of standard clinical management rather than exploring dedicated organ preservation strategies (planned intent). Only a small minority of locally advanced rectal cancers achieve a cCR with standard CRT; however, changing our focus to earlier cancer, which are more likely to respond to chemoradiotherapy, there is the

potential for considerably improving response rates. This opens for studies of dedicated and upfront non-surgical management strategies using chemoradiotherapy as a definitive treatment.

There is moderate evidence of a radiation dose–response relationship for pathological tumour regression after preoperative (chemo)radiotherapy.[15 16] Currently, there is no high-level evidence as to whether dose escalation increases the cCR rate, although selected publications have reported high cCR and organ preservation rates with high-dose chemoradiotherapy for early cancers.[17 18] Early response and toxicity data for a single-arm phase II organ preservation study delivering 62 Gy (tumour) and 50.4 Gy (lymph nodes) have also been presented in abstract form.[19] The non-randomised nature of these studies is a major limitation, particularly with the inclusion of small and early cancers which are more likely to respond.[20] Additionally, there is limited evidence of the optimum dose level for dose escalation, though experience from anal cancer and preliminary data from two ongoing Danish studies (NCT02438839, NCT04095299) indicate that 62 Gy can be safely delivered to a confined target volume in 2.2 Gy per fraction.

Modern radiotherapy techniques, including intensity-modulated radiotherapy (IMRT) and volumetric arc therapy (VMAT) with image guidance (image-guided radiotherapy, IGRT), improve conformality of treated volumes. Theoretically, this will reduce early and late toxicities when compared with conventional 3D conformal planning.[21] These advances offer the possibility of increasing the dose to the tumour without excessive morbidity of surrounding normal tissues. Dose escalation with simultaneous integrated boost can thus be achieved without significantly altering the dose delivered to the organs at risk, and without extending the overall treatment time.

To further minimise radiation-induced side effects, a risk-adapted target volume can be used. The risk of pelvic lymph node involvement or distal mesorectal nodal involvement is very low for patients with early rectal cancer.[22 23] Additionally, local recurrences are unlikely to be found above the level of the S2/3 interspace.[24] In early rectal cancer, it is thus doubtful that traditional, large elective treatment volumes are indicated. Therefore, it is reasonable to reduce the target volume to the peritumoral region of the primary tumour and the mesorectum.[25 26]

A Phase II trial of Higher RadiOtherapy Dose In The Eradication of early rectal cancer (APHRODITE) phase II study will use modern radiotherapy techniques to investigate the efficacy and safety of escalated dose radiotherapy for non-surgical treatment of early-stage rectal cancer in patients not suitable for standard surgical management.

## METHODS AND ANALYSIS
### Design and aim
APHRODITE is a phase II, multicentre, open-label randomised controlled trial of IMRT, with select use of

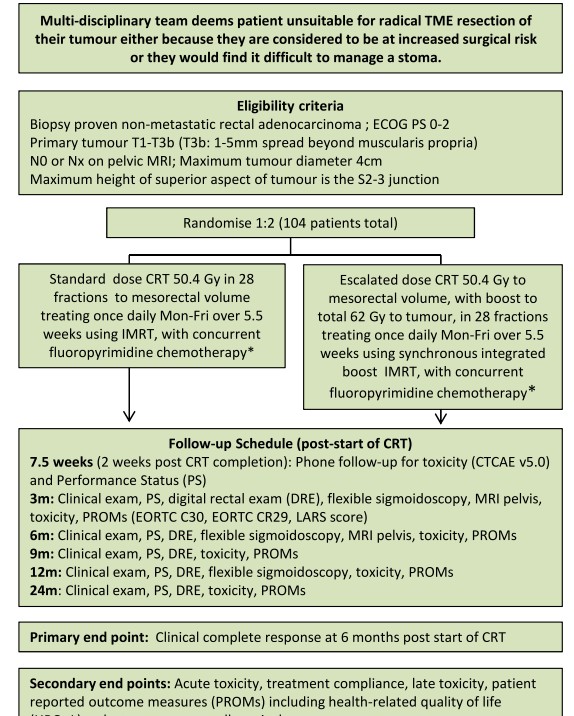

Multi-disciplinary team deems patient unsuitable for radical TME resection of their tumour either because they are considered to be at increased surgical risk or they would find it difficult to manage a stoma.

**Eligibility criteria**
Biopsy proven non-metastatic rectal adenocarcinoma ; ECOG PS 0-2
Primary tumour T1-T3b (T3b: 1-5mm spread beyond muscularis propria)
N0 or Nx on pelvic MRI; Maximum tumour diameter 4cm
Maximum height of superior aspect of tumour is the S2-3 junction

Randomise 1:2 (104 patients total)

Standard dose CRT 50.4 Gy in 28 fractions to mesorectal volume treating once daily Mon-Fri over 5.5 weeks using IMRT, with concurrent fluoropyrimidine chemotherapy*

Escalated dose CRT 50.4 Gy to mesorectal volume, with boost to total 62 Gy to tumour, in 28 fractions treating once daily Mon-Fri over 5.5 weeks using synchronous integrated boost IMRT, with concurrent fluoropyrimidine chemotherapy*

**Follow-up Schedule (post-start of CRT)**
**7.5 weeks** (2 weeks post CRT completion): Phone follow-up for toxicity (CTCAE v5.0) and Performance Status (PS)
**3m**: Clinical exam, PS, digital rectal exam (DRE), flexible sigmoidoscopy, MRI pelvis, toxicity, PROMs (EORTC C30, EORTC CR29, LARS score)
**6m**: Clinical exam, PS, DRE, flexible sigmoidoscopy, MRI pelvis, toxicity, PROMs
**9m**: Clinical exam, PS, DRE, toxicity, PROMs
**12m**: Clinical exam, PS, DRE, flexible sigmoidoscopy, toxicity, PROMs
**24m**: Clinical exam, PS, DRE, toxicity, PROMs

**Primary end point:** Clinical complete response at 6 months post start of CRT

**Secondary end points:** Acute toxicity, treatment compliance, late toxicity, patient reported outcome measures (PROMs) including health-related quality of life (HRQoL), colostomy rate, overall survival

**Figure 1** Trial schema. *At treating team discretion, concurrent chemotherapy can be used either at 75% dose or omitted completely. This choice must be declared prior to randomisation. CTCAE, Common Terminology Criteria for Adverse Events; ECOG, Eastern Cooperative Oncology Group; EORTC, European Organisation for Research and Treatment of Cancer; IMRT, intensity-modulated radiotherapy; LARS, low anterior resection syndrome; TME, total mesorectal excision.

concomitant chemotherapy, in patients with early-stage rectal cancer (T1-3bN0, maximum diameter ≤4 cm) who are deemed not suitable for radical TME surgery by their multidisciplinary team (MDT). The primary aim is to assess whether radiotherapy dose escalation increases the cCR rate at 6 months from the start of CRT, compared with standard radiotherapy dose CRT, with acceptable toxicity. Patients may receive full dose, reduced dose or no concomitant chemotherapy. A total of 104 eligible patients will be recruited from 10 to 12 UK radiotherapy sites. Participants will be randomised on a 2:1 basis to receive escalated dose or standard dose chemoradiation. The participant pathway can be seen schematically in figure 1. The study protocol and this manuscript have been written in accordance with the Standard Protocol Items: Recommendations for Interventional Trials guidelines.

## Trial objectives

The primary objective of APHRODITE is to compare the proportion of participants with a cCR at 6 months from the start of CRT. A composite definition for cCR is used, including digital rectal examination (DRE), rectal endoscopy and pelvic MRI, defined by:

► No evidence of either mucosal tumour or submucosal swelling on white light endoscopy. A flat white scar remains, with or without telangiectasia.
► No palpable tumour on DRE.
► High-resolution pelvic MRI scanning shows either a linear scar only or dense fibrosis with no obvious tumour signal (mrTRG 1 or 2).[27 28]

This definition of cCR follows international consensus.[29] Every effort will be made to ensure trimodality assessment (endoscopy, DRE and MRI). If such is not possible, confirmation should always include a rectal endoscopy and one other (DRE, MRI). Detailed guidance on MRI and endoscopy is provided in separate trial guidelines. The protocol does not require use of biopsy for response evaluation.

Currently research is unclear whether high-dose radiotherapy increases the rate of superficial mucosal ulcerations in the presence of a complete primary tumour response. Therefore, an alternative definition of the primary endpoint, with the inclusion of superficial mucosal ulceration classified as cCR, will be explored in a sensitivity analysis.

The secondary objectives of APHRODITE evaluate the safety and tolerability of dose escalation based on:
► Acute toxicities as per Common Terminology Criteria for Adverse Events (CTCAE V.5.0) (measured weekly during and up to 2 weeks after end of treatment).
► Treatment compliance (both radiotherapy and, if received, concomitant chemotherapy).
► Toxicities during follow-up as per CTCAE V.5.0 (measured up to the 24-month follow-up point).

The longer term activity of dose escalation will be evaluated based on:
► Time to stoma formation (time from randomisation to stoma formation—colostomy or ileostomy).
► Overall survival (time from randomisation to death from any cause).
► Patient-reported outcome measures (PROMs), assessed by the European Organisation for Research and Treatment of Cancer (EORTC) Quality of Life Questionnaires QLQ-C30[30] and QLQ-CR29,[31] low anterior resection syndrome (LARS) questionnaire[32] and additional EORTC QLQ items from the Item Library relevant for organ preservation.

## Study population

Eligible patients are those with early-stage rectal cancer who are deemed by their MDT not suitable for radical TME surgery because either:
► The patient is thought to be at increased surgical risk due to specific medical comorbidity or general frailty.
► The patient has marked anxiety at the prospect of having a stoma.
► It is anticipated that the patient would have difficulty managing a stoma postoperatively, for example, due to physical problems.

**Table 1** Inclusion and exclusion criteria

| Inclusion criteria | Exclusion criteria |
|---|---|
| 1. Biopsy-confirmed adenocarcinoma of the rectum. | 1. Nodal involvement on MRI (ie, N1–N2)* or discontinuous tumour deposits (N1c). |
| 2. Age 18 years or over. | 2. The presence of EMVI discontinuous with the primary tumour on MRI. |
| 3. Able to provide written informed consent. | 3. Involvement of anal intersphincteric plane or external anal sphincter or adjacent organs; or tumour involves/breaches levator ani. |
| 4. Patient deemed unsuitable for TME surgical resection, see main text for details. | 4. Tumour grown through and breached mesorectal fascia. |
| 5. Patient suitable for pelvic radiotherapy or chemoradiation: ECOG PS 0–2. | 5. Signet ring carcinoma or tumours histopathologically containing a neuroendocrine component; or dominant mucinous tumour on MRI. |
| 6. Primary tumour staged T1–T3b*, and maximum tumour diameter ≤4 cm, both on MRI. | 6. Undergone an attempt at complete local resection of their cancer. |
| 7. No unequivocally involved lymph nodes, NX and N0 eligible*. | 7. Definite distant metastases (equivocal distant metastases are permitted). |
| 8. Tumour visible on MRI. | 8. Defunctioning colostomy/ileostomy fashioned. |
| 9. Superior aspect of tumour is at or below a horizontal line drawn from the anterior aspect of the S2/3 junction on pretreatment MRI. | 9. Previous pelvic radiotherapy; or prior systemic chemotherapy for colorectal cancer. |
| 10. For low rectal tumours, superior to the puborectalis sling, the mesorectal fascia or levator is:<br>▶ Clear (>1 mm from disease to levator ani or mesorectal fascia).<br>▶ Or threatened (≤1 mm from disease to levator ani or mesorectal fascia).<br>▶ Or mesorectal fascia is involved but not breached. | 10. Prior invasive malignancy unless disease free for a minimum of 3 years (excluding basal cell carcinoma of the skin or other in situ carcinomas). |
| 11. Blood counts fulfilling:<br>▶ Estimated creatinine clearance ≥50 mL/min.<br>▶ Absolute neutrophil count >1.5×10$^9$/L; platelets >100×10$^9$/L.<br>▶ Serum transaminase concentration <3× upper limit normal (ULN), bilirubin concentration <1.5× ULN. | 11. Women who are pregnant, breast feeding or of childbearing potential and unwilling to use contraceptives. |

*Tumour, node, metastases (TNM) staging as per UICC 8th Edition.
ECOG, Eastern Cooperative Oncology Group; EMVI, extramural venous invasion; PS, Performance Status; TME, total mesorectal excision; UICC, Union for International Cancer Control.

See table 1 for a full list of inclusion and exclusion criteria.

## Randomisation and recruitment

A computer-generated minimisation programme incorporating a random element will be used to allocate patients on a 2:1 basis to receive escalated dose or standard dose chemoradiation, ensuring arms are well balanced for the following factors:

▶ Randomising site.
▶ T-stage (<T3 vs T3).
▶ Chemotherapy use (either 100% or 75% dose) versus no chemotherapy use.

Randomisation will be performed centrally via the University of Leeds Clinical Trials Research Unit (CTRU) automated 24-hour randomisation system. The registration and randomisation process will be instigated by on-site research staff; patient consent form must be attained prior to registration (see online supplemental file 1).

Recruitment will take place over 2 years, with an estimated monthly recruitment rate of four to six patients per month. APHRODITE opened in February 2020, with expected recruitment completion by winter 2022. The study has and will be presented at national and international meetings, increasing clinical awareness. A trial website and dedicated Twitter account will be used to further increase exposure, highlighting the trial population criteria to treating clinicians and MDT members. Based on preliminary feedback from clinicians and patient representatives in the study design phase, the 2:1 allocation between the intervention and control arms will make recruitment attractive for patients with limited standard treatment options.

## Sample size

The target sample size is 104 participants (70 experimental: 34 control), based on a two-group $\chi^2$ test of equal proportions, with continuity correction. Assuming a control arm cCR rate of 35%,[33] using a one-sided test at 20% significance, 104 participants (incorporating a 5% loss to follow-up) will provide 80% power to detect an absolute difference of 20% (to 55%) between arms. Data suggest potential differences as great as 25% (in favour of the experimental arm).[17 34]

An inflated type I error rate of 20% is used as the aim is to show preliminary evidence of activity.[35 36] The direct comparison with the control avoids the risk of incorrect decision-making due to uncertain historical control data.[37]

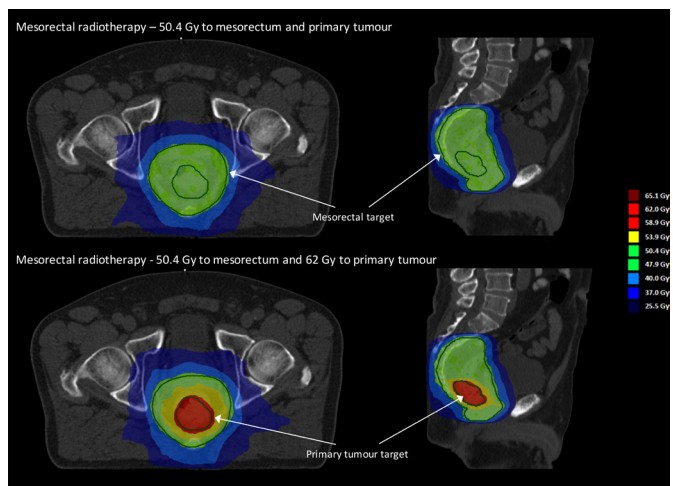

**Figure 2** Example treatment plans for the control arm (top panel: uniform 50.4 Gy in 28 fractions to the primary tumour and elective mesorectal volume) and intervention arm (lower panel: 62 Gy in 28 fractions to the primary tumour, 50.4 Gy to the elective mesorectal volume) for a single patient.

## Treatment regimen

Intervention arm: the primary tumour will receive 62 Gy in 28 fractions, with 50.4 Gy delivered to the surrounding elective mesorectal volume, using a simultaneously integrated boost.

Control arm: the primary tumour and elective mesorectal volume will receive a uniform 50.4 Gy in 28 fractions (preoperation standard dose).

A reduced mesorectum-only elective volume will be used, as appropriate for early-stage disease at low risk of nodal failure. All treatments will be planned and delivered using IMRT or VMAT, with extensive radiotherapy guidelines, based on the STAR-TREC (*S*ave the rectum by watchful waiting or *T*rans*A*nal microsurgery following (chemo) *RE*ctal *C*ancer) trial target definition and treatment planning.[25 26] Daily image guidance with cone beam CT will be mandatory and promotes precise treatment delivery, essential as the mesorectum and tumour exhibit significant changes in day-to-day position.[7 24] See figure 2 for an example treatment plan.

Concurrent chemotherapy will be used during CRT, either as single-agent oral capecitabine (825 mg/m$^2$) given two times per day from Monday to Friday on the days of radiotherapy throughout the course of radiotherapy, or alternatively as intravenous 5-fluorouracil/leucovorin delivered once per day (5FU 350 mg/m$^2$ plus leucovorin) concurrent with fractions 1–5 and 20–25 of radiotherapy (weeks 1 and 5). If the treating clinician decides a patient is not fit to receive the full (100%) dose of chemotherapy, due to comorbidities, or general frailty, they have the options to treat at 75% dose, or omit chemotherapy (radiotherapy alone). The intended chemotherapy treatment option must be declared for each patient *prior* to randomisation.

## Trial assessments and follow-up

All willing patients will be assessed prior to randomisation to confirm full eligibility (table 1). Recruited participants will be assessed at various timepoints (table 2).

For further detail of the participation flow, see trial schema (figure 1).

Adverse reactions (ARs), serious adverse reactions (SARs), related unexpected serious adverse events

**Table 2** Trial schedule of assessment

| | Early assessments | | | Treatment | | Follow-up assessment (measured post start of CRT) | | | | | |
| --- | --- | --- | --- | --- | --- | --- | --- | --- | --- | --- | --- |
| | Eligibility | Baseline | Pretreatment | On treatment | End of treatment | 2 weeks | 3 months | 6 months | 9 months | 12 months | 24 months |
| Informed consent | X | | | | | | | | | | |
| Histopathology | X | | | | | | | | | | |
| Performance status | X | | X | | X | X | X | X | X | X | X |
| Blood | X | | X | X | X | | | | | | |
| Pregnancy screening | X | | | | | | | | | | |
| ECG | | X | | | | | | | | | |
| MRI (pelvis) | X | | | | | | X | X | Local schedule | | |
| CT (chest, abdomen, pelvis) | X | | | | | | Local standard schedule | | | | |
| Flexible sigmoidoscopy* | X | | | | | | X | X | | X | |
| Digital rectal examination | X | | | | | | X | X | X | X | X |
| Clinical assessment | X | | | X | X | | X | X | X | X | X |
| Toxicity (CTCAE) | | X | | X | X | X | X | X | X | X | X |
| PROMs | | X | | | X | | X | X | X | X | X |

*Additional flexible sigmoidoscopy assessments may occur for monitoring according to local schedules
CRT, Chemoradiotherapy; CTCAE, Common Terminology Criteria for Adverse Events; PROM, patient-reported outcome measure.

(RUSAEs) and serious adverse events of interest (SAEoI) will be collected from randomisation until the 24-month follow-up point. SAEoIs are events identified by the research team that need additional monitoring, including angina/myocardial infarction, pulmonary embolism, serious skin reactions.

Data will be collected from all randomised participants, irrespective of treatment compliance. Where possible, endpoint and PROMs data will be collected from withdrawn participants who do not withdraw consent from further data collection.

### Treatment and response assessment quality assurance

The trial will implement robust quality assurance (QA) of radiotherapy treatment and the primary endpoint assessments. The radiotherapy QA programme will be implemented by the National Radiotherapy Trials Quality Assurance Group to ensure treatment is planned and delivered according to the trial protocol. All radiotherapy centres will complete pretreatment benchmark exercises. Anonymised pretreatment, 3-month and 6-month endoscopic photographs and MRI scans will be collected for all patients, with the first two patients from each site reviewed centrally. Trial Management Group (TMG) specialist members will ensure the inclusion/exclusion criteria have been met (baseline scan) and response has been correctly reported (3 and 6 months), providing feedback where necessary. Further review may be performed in batches.

### Statistical analysis

A full statistical analysis plan will be written before any analyses are undertaken. No formal interim analyses are planned within the trial.

The primary endpoint will be assessed on a *modified intention-to-treat* (*MITT*) *population*; all randomised participants who have had at least one dose of trial treatment, grouped by the treatment arm to which they were randomised (regardless of ineligibility, non-compliance or withdrawal).[38]

A multivariable logistic regression model, adjusted for the minimisation factors, will be used to compare the proportion of participants achieving a cCR at 6 months. Significance testing will be one sided at the 20% level. The treatment effect will be presented as an OR with a two-sided 60% CI (equivalent to a one-sided 80% lower confidence limit). Patterns of missing data and imputation methods (including multiple imputation) will be explored, if appropriate. A sensitivity analysis will be performed using the alternative definition of cCR.

All secondary analyses will use a 5% significance level and a two-sided CI, where appropriate. Safety data will be presented on the *safety population*; participants are summarised on the treatment received regardless of randomisation. All other data will be summarised on an MITT population.

Summary statistics will be presented for toxicities on the number and proportion of participants experiencing ARs, SARs, RUSAEs and SAEoI. Treatment compliance to the allocated radiotherapy (±chemotherapy) will be monitored and presented, including summaries for delays, dose modifications and discontinuation. PROMs will be summarised and reported using standard guidelines.[32 39 40] Repeated measures models will be used to assess differences between treatment arms. Cumulative incidence will be used to compare the time to stoma formation between treatment arms. Overall survival will be compared using the Cox proportional hazards model. Kaplan-Meier curves and survival estimates with corresponding CIs will also be presented.

### Trial organisation

Trial coordination, data management and statistical analysis will be directed and conducted by the trial-specific project team at the University of Leeds CTRU. Trial supervision will be established according to the principles of Good Clinical Practice and in line with the relevant Research Governance Framework within the UK and CTRU standard operating procedures.

### Data collection and management

Participating sites will record participant data on trial-specific case report forms and submit them to the CTRU. Participants will complete PROMs on the relevant paper questionnaire forms at the time of their clinic appointment. All trial data will be entered, validated and stored securely by CTRU. Missing and discrepant data will be flagged and additional data validations raised, as appropriate, from the CTRU data management team. Missing data (except PROMs) will be followed up until received, or confirmed as not available. Data items regarding consent, patient safety and the primary endpoint will be subject to manual priority checking.

Data will be stored securely at Leeds CTRU. Only CTRU will have access to the data, prior to analysis and release of trial results. On completion of the trial, data will be stored in the sponsor archiving facility for a minimum of 15 years. After the final trial results have been published, interested researchers may contact the TMG and CTRU to request access to relevant data. Any requests will be reviewed by the TMG.

### Trial monitoring

The TMG will provide ongoing clinical, practical and statistical advice on trial-related matters. The trial will be overseen by an independent Trial Steering Committee (TSC) and Data Monitoring and Ethics Committee (DMEC). The DMEC comprises two clinical oncologists and one statistician. They will review and monitor accumulating interim safety data and unblinded reports, at least annually. Their role is to protect the safety of the participants and maintain the research integrity of the study, advising the TSC on trial developments, including advice on trial continuation. A trial-specific DMEC charter has been developed in line with the Data Monitoring Committees: Lessons, Ethics Statistics Study Group

(DAMOCLES) recommendations.[41 42] The sponsor and TSC have ultimate oversight over the conduct and continuation of the trial.

## Patient and public involvement

APHRODITE has been developed with patient and public involvement (PPI) representatives from the earliest stages. Multiple aspects of both trial design and delivery have been shaped in collaboration with our PPI representatives, including the acceptability of the intervention, randomisation and the allocation ratio (2:1). One PPI representative was a coapplicant on the grant application (MJ). Two PPI representatives are members of the TMG (MJ, AB), and have provided patient perspective throughout the set-up and recruitment. On their suggestion, APHRODITE has a patient-facing trial website, and a patient newsletter will be issued at regular intervals, with updates of trial progress and results when available. Trial feedback and communication with participants has been highlighted as a key importance by our PPI representatives.

## Patient preference substudy

Dose escalation is associated with a greater probability of tumour control at the cost of greater risk of toxicity, yet little information exists as to what risks patients are willing to accept for a better chance of tumour control. The APHRODITE patient preference study (chief investigator EW), to be conducted alongside the main trial, aims to address this knowledge gap.

The patient preference substudy uses a well-established survey technique called a discrete choice experiment (DCE).[43] DCEs have a simple response format in which patients make a series of choices between hypothetical treatments characterised by attributes, such as probability of tumour control after 2 years and the risk of side effects.[44] The attribute levels change in each question, allowing the trade-offs participants make to be analysed.[45] The survey has been developed using qualitative methods in line with good practice recommendations.[46] Analysis of responses will quantify maximum acceptable risk of toxicity for a given probability of tumour control, as well as their preferences for process attributes, such as the number of treatment sessions and after-treatment support. Participants will complete the survey before treatment and 6 months after treatment. This will allow us to assess whether preferences are systematically influenced by treatment experience.

The substudy plans to recruit 100–200 participants from sites participating in the APHRODITE trial, including APHRODITE participants, patients who decline APHRODITE participation, in addition to patients who are being managed non-surgically but do not meet the trial inclusion/exclusion criteria. Separate ethical approval has been obtained for the patient preference study from a National Health Service (NHS) Research Ethics Committee (reference 19/NE/0249).

## Sample collection

Pretreatment routine diagnostic formalin-fixed paraffin-embedded tumour biopsy tissue samples will be collected on all patients and sent to the University of Leeds for storage. Consent will be taken for their use in future translational research. The routine glass H&E stained slides used for diagnosis will be scanned to create a digital pathology resource for QA of the trial and support translational research, including research examining biomarkers of radiosensitivity and radioresistance.

## ETHICS AND DISSEMINATION

The trial obtained ethical approval from the North West Greater Manchester East Research Ethics Committee (reference number 19/NW/0565) and is registered in the ISRCTN registry. Confirmed by the UK Medicines and Healthcare products Regulatory Agency, APHRODITE is not a clinical trial of an investigational medicinal product (non-CTIMP). The trial is currently adhering to protocol version 3.0 (approved 12 May 2021).

The primary endpoint and early toxicity data will be analysed and reported when the last participant reaches 6 months after start of treatment. Further analyses will be conducted when all patients have reached 24 months of follow-up. The final trial publication will be written by the TMG, and will adhere to International Committee of Medical Journal Editors guidelines.

## DISCUSSION

There is currently a lack of prospective trials of dedicated non-surgical management strategies in rectal cancer. It is therefore unclear whether increased radiotherapy dose may contribute to short-term tumour response, long-term local control and organ preservation. In particular, the lack of randomised data from the early-phase setting has made it difficult to elucidate whether the high response rates observed in phase II trials are due to patient selection or treatment. The APHRODITE trial will provide high-level evidence whether dose escalation increases the cCR rate when directly compared with standard dose. Further studies will be required to provide evidence for whether a higher radiotherapy dose can provide improved long-term local control without surgery.

Other ongoing studies are examining dedicated organ preservation strategies for early rectal cancer. The STAR-TREC phase III trial (ISRCTN14240288) randomises patients with early (T1-3bN0) cancer with a preference for organ preservation between short-course and long-course standard (chemo)radiotherapies, with the aim of non-surgical management or local excision only. Both trials remain complimentary: STAR-TREC uses standard radiotherapy in a patient population suitable for surgery, while APHRODITE will assess intensified treatment in an alternative patient group. Three other studies of radiotherapy dose escalation strategies for organ preservation are ongoing: the international phase III OPERA trial

(Organ Preservation in Early Rectal Adenocarcinoma, NCT02505750) randomises patients between moderately dose-escalated CRT (54 Gy) and standard dose CRT (45 Gy), followed by Papillon contact X-ray therapy (90 Gy in three fractions); the MORPHEUS trial (More Organ Preservation: High dose rate brachytherapy versus External beam radiation therapy Multicenter Study - A randomised phase III trial, NCT03051464) randomises patients between moderately dose-escalated CRT (54 Gy) and CRT (45 Gy), followed by brachytherapy (30 Gy in three fractions); and the Danish multicentre WW3 phase II trial (NCT04095299) randomises patients between dose-escalated CRT (62 Gy) and standard dose CRT (50.4 Gy). All these studies require suitability for surgery, and the first two depend on access to specialised treatment equipment. The APHRODITE study thus fills a clear research gap in the current international trial portfolio.

Studies of non-surgical management and organ preservation for rectal cancer have focused on clinical endpoints, for example, cCR rate, stoma rate and overall survival, as well as the clinicians' interpretation of patient outcome. There is a paucity of toxicity data including PROMs following non-surgical management, particularly in a patient population usually excluded from clinical trials. The APHRODITE study will prospectively measure toxicity and PROMs, providing a valuable and unique data set.

At present, there is a lack of understanding of how patients weigh and prioritise different potential outcomes, including the balance between efficacy and toxicity, in the setting of non-surgical rectal cancer treatment. The decision regarding which severity of treatment-related toxicities is acceptable for a given organ preservation strategy has previously been evaluated purely by clinicians. Two small studies examined patient preferences for organ-preserving and surgical treatment approaches.[47 48] However, no studies have focused specifically on patient attitudes to treatment characteristics and outcomes following non-surgical management. The APHRODITE patient preference substudy will help inform future dose escalation studies and help non-surgical management of rectal cancer to better reflect patient perspectives.

The APHRODITE trial will provide valuable, high-quality, patient-centred evidence for escalated dose radiotherapy in patients with rectal cancer not suitable for radical surgery, addressing an important research gap.

**Author affiliations**
[1]Leeds Institute of Clinical Trials Research, University of Leeds, Leeds, UK
[2]Department of Radiology, Wrexham Maelor Hospital, Wrexham, UK
[3]Academic Department of Surgery, Queen Elizabeth Hospital Birmingham, Birmingham, UK
[4]Leeds Cancer Centre, St James's University Hospital, Leeds, UK
[5]Patient Advocate, Conwy, Wales, UK
[6]Leeds Institute of Medical Research at St James's, University of Leeds, Leeds, UK
[7]Medical Physics and Biomedical Engineering, University College London, London, UK
[8]National Radiotherapy Trials QA (RTTQA) Group, Mount Vernon Cancer Centre, Northwood, UK
[9]Patient Advocate, London, UK
[10]Department of Radiology, The Christie NHS Foundation Trust, Manchester, UK
[11]Department of Clinical Oncology, The Christie NHS Foundation Trust, Manchester, UK
[12]Academic Unit of Health Economics, Leeds Institute of Health Sciences, University of Leeds, Leeds, UK
[13]North Wales Cancer Treatment Centre, Glan Clwyd Hospital, Bodelwyddan, UK

**Contributors** Contributors to the conception and design: AA, DS-M, SB, SG. Development of the protocol, trial documents and patient information sheet: AA, AB, AG, AS, AW, CB, DS-M, DH, EMH, JS, MJ, MS, NW, RA, RK, SB, SG, SN. Writing the manuscript: EMH, AA. Reviewed the manuscript: AB, AG, AS, AW, CB, DS-M, EW, MH, MS, MT, NW, SB, SPB, SG. All authors read and approved the final manuscript. SG and AA share last authorship.

**Funding** APHRODITE is supported by Yorkshire Cancer Research (award number L411). The study is coordinated by Leeds CTRU and sponsored by the University of Leeds (GA18/118305). For inquiries of the trial, email APHRODITE@leeds.ac.uk. Trial website: https://ctru.leeds.ac.uk/aphrodite/

**Disclaimer** Both the sponsor and the funder have no contribution to the study design and concept; collection, management, analysis and interpretation of data; and the writing of and decision to submit reports for publication. The sponsor and the funder will review all the manuscripts prior to publication.

**Competing interests** EW reports grants from Yorkshire Cancer Research during the conduct of the study. MS reports personal fees from Servier, personal fees from Amgen, personal fees from Merck, outside the submitted work. NW reports grants from Yorkshire Cancer Research during the conduct of the study; grants from Cancer Research UK, outside the submitted work.

**Patient and public involvement** Patients and/or the public were involved in the design, or conduct, or reporting, or dissemination plans of this research. Refer to the Methods section for further details.

**Patient consent for publication** Not required.

**Provenance and peer review** Not commissioned; externally peer reviewed.

**ORCID iDs**
Eleanor M Hudson http://orcid.org/0000-0001-8758-7163
Sarah Brown http://orcid.org/0000-0002-7975-7537
Maria Hawkins http://orcid.org/0000-0002-6669-0628
Edward JD Webb http://orcid.org/0000-0001-7918-839X

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
