## [Reviewer comments · BMJ Open]

ARTICLE DETAILS

TITLE (PROVISIONAL)	A Phase II trial of Higher Radiotherapy Dose In The Eradication of early rectal cancer (APHRODITE): Protocol for a multicentre open label randomised controlled trial
AUTHORS	Hudson, Eleanor; Nutch, Samantha; Brown, Sarah; Adapala, Ravi; Bach, Simon; Burnett, Carole; Burrage, Alwyn; Gilbert, Alexandra; Hawkins, Maria; Howard, Debra; Jefford, Monica; Kochhar, Rohit; Saunders, Mark; Seligmann, Jenny; Smith, Alexandra; Teo, Mark; Webb, Edward; Webster, Amanda; West, Nicholas; Sebag-Montefiore, David; Gollins, Simon; Appelt, Ane

VERSION 1 – REVIEW

REVIEWER	Fleshman, James Baylor University Medical Center at Dallas
REVIEW RETURNED	19-Feb-2021

GENERAL COMMENTS	Please add the following to the exclusion criteria; Tumor greater than 4cm in maximum dimension
---

REVIEWER	Grabenbauer, Gerhard Coburg Cancer Centre, Radiation Oncology
REVIEW RETURNED	01-Mar-2021

GENERAL COMMENTS	Clearly written protocol, albeit concomitant chemotherapy is not well defined and chemotherapy according to O'Connell obviously not allowed!?
---

REVIEWER	Ikematsu, Hiroaki National Cancer Center Hospital East, Department of Gastroenterology & Gastrointestinal Oncology
REVIEW RETURNED	05-Jul-2021

GENERAL COMMENTS	This was a protocol of the study evaluating whether radiotherapy dose (62Gy) escalation increases the cCR rate, compared with standard radiotherapy dose CRT (50.4Gy), with acceptable toxicity. I think this study is very interesting, however, there are some problems. My main issues are as follows. Major 1) Were there historical control data of the control arm therapy, including efficacy and safety? Please show.2) In addition, were there the previous report about the data of the intervention arm therapy?3) How much do you think the intervention arm treatment will be more toxic than the control arm treatment? This is the very important information you need to think about the risk-benefit balance. Please explain.
--

	4) It is often difficult to distinguish residual lesions from post-treatment scars. Why did not require use of biopsy in evaluating cCR, which was the primary endpoint. I also think the details of the clinical diagnostic criteria for cCR should be specified. This seem affect the result of this study. 5) How do you evaluate the pretreatment T stage diagnosis? Since there are differences in lymph node metastasis rate and prognosis at each T stage, there may be differences in prognosis after CRT. Minor 1) In this study, the primary endpoint was cCR. Was cCR rate regarded as surrogate marker of prognosis for this study's population? 2) This study seemed to adopt a phase2 screen design, is that correct? Please specify if it matches. 3) Are there criteria for deciding whether or not to use chemotherapy?
--	--

VERSION 1 – AUTHOR RESPONSE

Reviewer: 1

Dr. James Fleshman, Baylor University Medical Center at Dallas Comments to the Author:

Please add the following to the exclusion criteria; Tumor greater than 4cm in maximum dimension

The inclusion criteria 'maximum tumour diameter ≤ 4 cm' is listed within table 1. In order to keep a succinct list of inclusion/ exclusion criteria we have chosen not included converse statements for all criteria (Page 12 Tracked).

Reviewer: 2

Prof. Gerhard Grabenbauer, Coburg Cancer Centre Comments to the Author:

Clearly written protocol, albeit concomitant chemotherapy is not well defined and chemotherapy according to O'Connell obviously not allowed!?

Additional detail has been added for clarity on use of chemotherapy to include dose (Page 14 Tracked). Use of additional oxaliplatin in this setting would not be regarded as standard of care in UK:

Concurrent chemotherapy will be used during CRT, either as single agent oral capecitabine (825 mg/m²) given twice per day Monday-Friday on the days of radiotherapy throughout the course of radiotherapy, or alternatively as intravenous 5-Fluorouracil/Leucovorin delivered once per day (5FU 350mg/m² plus Leucovorin) concurrent with fractions 1-5 and 20-25 of radiotherapy (weeks 1 and 5).

Reviewer: 3

Dr. Hiroaki Ikematsu, National Cancer Center Hospital East Comments to the Author:

This was a protocol of the study evaluating whether radiotherapy dose (62Gy) escalation increases the cCR rate, compared with standard radiotherapy dose CRT (50.4Gy), with acceptable toxicity. I think this study is very interesting, however, there are some problems. My main issues are as follows.

Major

1) Were there historical control data of the control arm therapy, including efficacy and safety? Please show.

Information on the safety of the control arm is present in the background: The option of offering non-surgical management to patients with cCR has shown to be safe [11-13] specifically for locally advanced rectal cancer patients (page 7 Tracked). Within the sample size paragraph (page 13/ 14

Tracked) we reference an assumed a control arm cCR rate of 35% (Dossa 2017) [33], representing the efficacy based on a review of studies for control data taking into account the T stages included in this study. We would prefer to keep the safety and efficacy within these two sections as we feel this best fits the paper structure and avoids duplication.

2) In addition, were there the previous report about the data of the intervention arm therapy? Within the introduction and background, we referenced data on high dose radiotherapy 'selected publications have reported high cCR and organ preservation rates with high-dose chemoradiotherapy for early cancers [17].' However, these were non-randomised studies and therefore support the need for a randomised control trial. We have now added a reference for the long-term outcomes of this study (Dizdarevic 2019)[18] and an additional reference for a watch and wait study reporting early toxicity and response data for high-dose radiotherapy using the same dose and fractionation as the current trial (Jensen 2020) [19]. (Page 7 Tracked)

3) How much do you think the intervention arm treatment will be more toxic than the control arm treatment? This is the very important information you need to think about the risk-benefit balance. Please explain.

There is currently no randomised, comparative data for the control and intervention arm, therefore the level of additional toxicity for the intervention is difficult to distinguish. The APHRODITE trial is a phase II study and has a focus on acute & late toxicity endpoints and patient reported outcome measures, with adverse reactions and safety monitored by a data monitoring committee throughout the study. This trial thus aims to provide the data to answer this very question.

4) It is often difficult to distinguish residual lesions from post-treatment scars. Why did not require use of biopsy in evaluating cCR, which was the primary endpoint. I also think the details of the clinical diagnostic criteria for cCR should be specified. This seem affect the result of this study.

The role of biopsies in evaluating cCR has previously been a key topic of discussion in the rectal cancer organ preservation field, however, a recent consensus was reached that best practice is to not rely on biopsy for confirm cCR, as biopsy has no additional diagnostic value (Fokas 2021).

The consensus publication referenced above specifically states "Another important point concerns the role of biopsy sampling in patients with a ncCR [near-complete clinical response] or cCR. In both scenarios, consensus agreement was reached that biopsy sampling does not provide additional value and could lead to false-negative results. Long-term follow-up data from a prospective study assessing the watch-and-wait strategy after CRT in this setting indicate that biopsies have only limited clinical value for ruling out residual cancer. A further analysis of data from this study clearly indicates that biopsy samples provide no added diagnostic value, especially when the criteria for a cCR are fulfilled. (...) the panel did not recommend a biopsy as mandatory for ncCR, for the abovementioned reasons."

The consensus reference has now been added to the main text (page 9):

This definition of cCR follows international consensus [29].

5) How do you evaluate the pretreatment T stage diagnosis? Since there are differences in lymph node metastasis rate and prognosis at each T stage, there may be differences in prognosis after CRT.

Evaluating baseline T stage is based on standard diagnostic methods in the UK including diagnostic MRI scans. Additional clarity has been added to Table 1 inclusion criteria 6 where primary tumour

stage and tumour diameter will both be assessed by MRI. We recognise the potential for T stage to effect prognosis after CRT and have included this as a minimisation factor within the randomisation and will adjust for this in the final analysis. This information is already contained in the “Randomisation and recruitment” section.

Minor

1) In this study, the primary endpoint was cCR. Was cCR rate regarded as surrogate marker of prognosis for this study’s population?

The primary endpoint cCR is not currently a validated surrogate endpoint for long-term tumour control without surgery, when considering guidance such as the Prentice criteria (Prentice 1989). However, it is recognised as a useful endpoint in earlier phase studies focused on assessing the potential of novel treatment strategies to increase the rate of organ preservation (Fokas 2021). Specifically the consensus paper states “Early assessments of tumour response (such as the cCR rate) should be used as primary end points for early phase I/II trials designed to identify strategies that increase cCR rates and enable NOM or LE using more intensive radiotherapy, CRT or total neoadjuvant treatment (TNT) regimens to select tolerable and locally effective treatment regimens for further testing in larger cohorts, such as the Danish trial⁷ or the recently completed CAO/ARO/AIO-16 trial (NCT03561142). Notwithstanding, both the risks and the benefits of treatment intensification should be considered carefully in these contexts.”

2) This study seemed to adopt a phase2 screen design, is that correct? Please specify if it matches.

The term ‘screening design’ is used for different settings in phase II, with many using this to describe a multi-arm phase II trial with the aim of screening multiple agents for selection. Our trial is rather a phase IIb trial to evaluate dose escalation compared to standard dose radiotherapy only, to determine whether there is sufficient evidence to move forward to larger scale trials. While the design fits that described by Rubinstein and Stone, given the potential for confusion with multi-arm trials we felt it was best to not update the description of the study.

3) Are there criteria for deciding whether or not to use chemotherapy?

The decision to treat with chemotherapy is made prior to randomisation and on a patient-by-patient basis by the treating clinician. This was a pragmatic decision and was chosen to best reflect the standard practice in this patient group; therefore, we have not included any decision criteria for this element of treatment. Factors such as general patient fitness or specific contra-indications to the use of concurrent chemotherapy, will dictate this decision.

VERSION 2 – REVIEW

REVIEWER	Ikematsu, Hiroaki National Cancer Center Hospital East, Department of Gastroenterology & Gastrointestinal Oncology
REVIEW RETURNED	27-Sep-2021
GENERAL COMMENTS	The authors responded appropriately to the questions. I have no further questions or comments.